# In-Situ Al-Mg Alloy Base Composite Reinforced by Oxides and Intermetallic Compounds Resulted from Decomposition of ZrW_2_O_8_ during Multipass Friction Stir Processing

**DOI:** 10.3390/ma16020817

**Published:** 2023-01-14

**Authors:** Andrey Chumaevskii, Anna Zykova, Alexandr Sudarikov, Evgeny Knyazhev, Nickolai Savchenko, Alexander Gubanov, Evgeny Moskvichev, Denis Gurianov, Aleksandra Nikolaeva, Andrey Vorontsov, Evgeny Kolubaev, Sergei Tarasov

**Affiliations:** 1Institute of Strength Physics and Materials Science, Siberian Branch of Russian Academy of Sciences, 634055 Tomsk, Russia; 2Institute of Inorganic Chemistry, Siberian Branch of Russian Academy of Sciences, 630090 Novosibirsk, Russia

**Keywords:** ZrW_2_O_8_, negative thermal expansion, aluminum matrix composites, friction stir processing

## Abstract

In the presented work, the effect of friction stir processing admixing the zirconium tungstate ZrW_2_O_8_ powder on the microstructure, mechanical and tribological properties of the AA5056 Al-Mg alloy stir zone has been studied. The FSP resulted in obtaining dense composite stir zones where α-ZrW_2_O_8_ underwent the following changes: (i) high-temperature transformation into metastable β’-ZrW_2_O_8_ and (ii) decomposition into WO_3_ and ZrO_2_ oxides followed by the formation of intermetallic compounds WAl_12_ and ZrA_l3_. These precipitates served as reinforcing phases to improve mechanical and tribological characteristics of the obtained fine-grained composites. The reduced values of wear rate and friction coefficient are due to the combined action the Hall–Petch mechanism and reinforcement by the decomposition products, including Al_2_O_3_, ZrO_2_, β’-ZrW_2_O_8_ and intermetallic compounds such as WAl_12_ and ZrAl_3_. Potential applications of the above-discussed composites maybe related to their improved tribological characteristics, for example in aerospace and vehicle-building industries.

## 1. Introduction

Aluminum matrix composites (AMCs) are considered a new emerging class of materials with properties to be tailored to specific applications [1,2,3,4,5]. Aluminum has low density, good processability and heat capacity. However, its high coefficient of thermal expansion (CTE) (22.4 × 10^−6^ K^−1^) limits its application in the field of microelectronics and precision instruments. Novel advanced composites have been developed that consist of particles with negative thermal expansion (NTE) and metal matrices and exhibit low thermal expansion with excellent thermal stability and conductivity [6,7,8,9,10]. These properties are well-suited for the precision applications where minimal thermal expansion is needed. The iron tungstate ZrW_2_O_8_ is an excellent isotropic material that demonstrates negative thermal expansion (NTE) over a wide temperature range (273–777 °C), so it may be an ideal component for preparing the aluminum matrix composite [6,7,8,9,10].

The ZrW_2_O_8_ has three phases: α-phase is stable at the room temperatures, β-phase is stable at high temperatures and γ-phase is stable at high pressure. The α-phase with CTE = −8.7 × 10^−6^ K^−1^ undergoes a reversible transformation into the β-phase with CTE = −4.9 × 10^−6^ K^−1^ at 155 °C. When exposed to pressure above 0.2 GPa, the cubic α-phase is transformed into the high-pressure orthorhombic γ-phase with a significantly higher CTE value in the range −8.7 to −1.0 × 10^−6^ K^−1^. When heated to 120 °C the γ-ZrW_2_O_8_ may turn again into the α-ZrW_2_O_8_.

The most important disadvantage of the ZrW_2_O_8_ is that it is metastable and decomposes into ZrO_2_ and WO_3_ when heated above 770 °C in air [6,7,8,9,10,11]. In addition, it has been shown in several publications that the decomposition temperature of ZrW_2_O_8_ in an aluminum matrix drops to about 410 °C [6,9,10]. At the same time, it was shown [11] that the use of zirconium tungstate powder as an additive for reinforcing an aluminum alloy A356 led to increasing both hardness and strength of such a composite, as well as to improving its fatigue life.

The formation of a composite structure in an aluminum alloy can be achieved by different methods: casting with stirring [12,13,14,15], powder metallurgy [16], laser surface alloying [17,18], electromechanical machining [19,20], friction surfacing [21], lateral friction surfacing [22] and friction stir processing (FSP) [23,24,25,26]. It was mentioned above that using fusion processes such as casting or laser melting would definitely decompose the zirconium tungstate, especially in the presence of such a strong reducing agent as aluminum.

The powder metallurgy approach showed that ZrW_2_O_8_ decomposed during sintering at 600 °C [16] and there is still a need to reduce the treatment temperatures, which is possible with the use of solid-state processes such as, for example, friction-base ones.

The friction-base processes such as friction surfacing and lateral friction surfacing use consumable rods and may be used for covering an alloy with a dissimilar metal. The lateral friction surfacing is a novel method that allows for saving the rod metal and reducing the process temperature [27]. These processes may be used for fabricating composite coatings intended for resisting wear and/or corrosion [28,29]. Despite using preheating and claiming the strong bonding between the coating and the substrate metal, all macrographs allow observing a sharp boundary between the coating and substrate, which is a stress concentrator and therefore should be avoided. When it comes to preparing the composites only limited methods can be used, for example, depositing powder bed on the substrate with the following friction surfacing will not definitely provide good bonding.

Friction stir processing is a very promising technology for formation of a composite structure in the subsurface layer of a metal in the solid state because instead of bonding the consumable rod metal to the substrate it allows for obtaining fully intermixed stir zone. The friction stir processing makes it possible to produce local hardening of the metal with a minimum thermal effect [24]. Improving the mechanical and tribological properties of the material surface is possible with the introduction of various ultrafine powders of refractory materials [24]. At the moment, there are no published data on the effect of ZrW_2_O_8_ on aluminum alloys after FSP.

In the present work, the effect of FSP admixing of the ZrW_2_O_8_ powder into AA5056 on stir zone microstructure as well as mechanical and tribological properties of the stir zone metal.

## 2. Materials and Methods

The micron-sized ceramic particles of ZrW_2_O_8_ (~3 μm) were obtained by courtesy of the Institute of Inorganic Chemistry, Siberian Branch of Russian Academy of Sciences, Novosibirsk, Russia. These ZrW_2_O_8_ powders were produced by thermal decomposition at 570 °C of the precursor ZrW_2_O_7_ (OH)_2_·2H_2_O synthesized under the hydrothermal synthesis conditions [30].

To carry out the FSP on the AA5056, an experimental FSP machine developed at the Institute of Strength Physics and Materials Science, Siberian Branch, Russian Academy of Sciences, has been used. The substrate materials were AA5056 Al-Mg alloy plates with 300 (L) × 70 (W) × 5 (D) mm^3^ dimensions. A row of holes was drilled in these plates which then were filled with the ZrW_2_O_8_ powder (Figure 1a). The total powder volume content was calculated per a volume of FSPed stir zone (SZ) and totaled to 10 vol.%. Then, the powder was mechanically compacted in the holes. After that, the plate was subjected to friction stir processing.

The friction stir processing was carried out using eight successive passes of the tool. This FSP tool (ISPMS SB RAS, Tomsk, Russia) with a truncated cone 3 mm height pin, ∅20 mm shoulder, and inclination angle of 3° was machined from a high-quality heat-resistant H13 steel. A water-cooling system was used to avoid overheating the FSP tool during processing. Argon shielding served to protect the processed metal against oxidation. For uniform and defect-free processing, the following FSP parameters were used: axial force on the tool 1300 kg, tool rotation frequency 500 rpm, tool speed 90 mm/min. The choice of the FSP parameters was based on our previous experimenting with friction stir welding and processing on the Al-Mg alloys. This set of parameters allowed obtaining defectless stir zones and micron-sized grains that provided higher than 100% welded joint strength in comparison with that of the as-received hot-rolled AA5056.

The friction stir processing is shown in Figure 1b. The fabricated surface composites were denoted as ZrW_2_O_8_/AA5056. A sample which was subjected to FSP with no reinforcement was named as FSPed base metal.

The samples were polished with sandpapers of 600 to 2000 grit. To polish the sections, the diamond paste ASM 2/1 NOMG was used, evenly applied in a thin layer on the fabric surface. Etching was carried out in Keller’s reagent by immersing the thin section for 15–20 s in 0.5 mL of the solution.

Microhardness was measured using an AFFRI DM8 microhardness tester (*AFFRI*, Induno Olona, Italy) with an indenter load of 50 g with an inter-indentation step of 0.5 mm. The determination of the ultimate tensile strength of the samples was carried out during mechanical tensile machine UTC-110M-100 (Testsystems, Ivanovo, Russia).

The pin-on-disk sliding using a TRIBOtechnic tribometer (TRIBOtechnic, Clichy, France) was applied for measuring the friction and wear characteristics of samples. Samples in the form of 3 × 3 × 10 mm^3^ pins were cut off the ZrW_2_O_8_/AA5056 stir zone, while Ø35 mm, 5 mm thickness counterbody disks were machined from a quenched AISI 420 steel rod. The rotation rate, normal load and sliding path length were 94 RPM, 15 N and 5600 m, respectively.

A symmetrical Bragg–Brentano XRD configuration (θ/2θ) with the Co-Kα radiation was carried out to identify phases formed in the surfaces of the FSPed composites. Grazing-incidence X-ray diffraction (GIAXRD) with the Co-Kα radiation was used to detect phases in the subsurface on the worn surfaces of the FSPed composites at beam incidence angle of 10°. These experiments were carried out using a DRON-7 X-ray diffractometer (Burevestnik, Russia) with a scan range of 15–80° with (2θ) step size of 0.05°. Identification of the XRD reflections was performed using Crystal Impact’s software “Match!” (Version 3.9, Crystal Impact, Bonn, Germany).

The microstructure of the FSPed composites after the preparation process was studied using an Olympus OLS LEXT 4100 laser scanning microscope (Olympus Corp., Tokyo, Japan), an Altami Met 1S optical microscope (Altami Ltd., Saint-Petersburg, Russia) and an X-ray computer tomography YXLON Cheetah EVO (YXLON International GmbH, Hamburg, Germany).

To study the elemental composition and microstructure of the FSPed composites after the preparation process and after the friction tests, a scanning electron microscope (SEM) TESCAN VEGA 3 SBU (TESCAN ORSAY HOLDING, Brno, Czech Republic) equipped with electron energy dispersive spectroscopes (EDS) OXFORD X-Max 50 (Oxford Instruments, Concord, MA, USA) with 20 kV accelerating voltage, 4–12 nA current, and approximately 2 µm probe spot diameter has been used. The fine structure and phase composition of the samples were studied using a JEM-2100 transmission electron microscope (JEOL Ltd., Tokyo, Japan). The thin foils for TEM were prepared via the ionic thinning method using an EM-09100IS machine (JEOL Ltd., Tokyo, Japan).

## 3. Results

### 3.1. Microstructure and Phase Compound of ZrW_2_O_8_ Reinforcement Particles

As shown in Figure 2a, the ZrW_2_O_8_ particles were identified as the α-ZrW_2_O_8_ ones. According to both SEM and TEM images, the zirconium tungstate powder is represented by the intergrown or isolated high aspect ratio particles (Figure 2b,c). The SAED pattern in Figure 2c shows numerous reflections arranged in rings, which reflect the polycrystalline structure of the particles.

### 3.2. Macrostructure and Microstructure Evaluations of FSP-ed Composites

The FSP process intensity as well as the plasticized metal response can be evaluated by parameters such as tool rotation torque, mechanical reaction force and surface temperatures. The reaction force is the resistance of the metal with respect to traverse movement of the tool along the FSP track, which is measured using a gauge on the FSP tool shaft. In order to evaluate the heat input, both mechanical parameters were converted in corresponding power contributions and the plotted vs. the FSP pass times (Figure 3a,b). The contribution of power from the tool rotation is about two orders of magnitude higher than that of the reaction force. The power contribution from the torque is almost the same for the first and fourth FSP passes while for the eighths one it is a bit lower (Figure 3a). The reaction force power shows some tendency for reducing the pass number but there is not much difference between the fourth and eighth passes (Figure 3b). The temperature measured by an IR thermal imager reached the maximum value of about 450 °C for composites after eight tool passes (Figure 3c). The fact that the surface temperatures achieved during FSP on the base metal were always lower than those obtained during the intermixing may be evidence in favor of additional heat release due to exothermic reactions occurring in the SZ during stirring. In the general FSP case, these reactions are related to precipitation of intermetallic compounds.

It follows from the experimental results presented below that the α-ZrW_2_O_8_ particles underwent a polymorphic transformation during the preparation of FSP-ed composites and then participated in diffusion reactions, the products of which are intermetallic compounds and oxides, henceforth called as “reaction product particles” (RPPs), irrespective of the specific composition.

The FSPed stir metal macrostructure is typical for FSP that consists of the center stir zone (Figure 4, SZ), advancing side (Figure 4, AS) and retreating side (Figure 4, RS). Figure 4a–d shows metallographic images of AA5056 SZ sections with mixed RPPs after one to four successive tool passes. It is clearly seen that after one pass, the mechanically compacted powder had no time to be homogeneously distributed throughout the entire SZ volume, and agglomerated in a black on the advancing side (Figure 4a). After the second and third passes (Figure 4b,c) the agglomerate zone area reduced and more homogeneous distributions were obtained due to the tool-driven flow of the plasticized metal.

On the fourth pass, the agglomerate disappeared and more or less homogeneous distribution of the RPPs was achieved in the bulk of the SZ with the only RPPs agglomeration in the form of a subsurface layer created by the shoulder-driven flow (Figure 4d).

To avoid such a detrimental effect and thus provide more homogeneous distribution of RPPs in the SZ, another one to four FSP passes were performed with parameters as follows: 1300 kg axial load, tool traverse speed 900 mm/min and tool rotation rate 500 RPM (Figure 5a–d). It is noticeable that after the additional passes, the mixing zone became more pronounced and contrasting as well as acquired its characteristic annular shape. It can be seen that after the single additional pass (Figure 5a), a part of the RPPs were displaced closer to the surface. After the second and third additional passes (Figure 5b,c) no noticeable changes in the stir zone are observed, the remaining RPPs in the treatment area are evenly distributed throughout the SZ volume. On the fourth additional pass, RPPs are evenly distributed throughout the SZ, with some of it being concentrated in the subsurface layer. No visible defects (voids, cracks, tunnels) were found after all additional passes. Thus, defect-free surface formation became possible due to correctly selected process parameters.

The above-discussed macroscale inhomogeneity of the RPPs distribution has been studied using the X-ray computer tomography (XCT) (Figure 6). The XCT images of the RPPs/AA5056 composite show that after the first pass the RPPs concentrated on the retreating sides mainly in the form of large agglomerates (Figure 6a,b). After the fourth pass the RPP distribution pattern is changed so that some RPPs are left on the advancing side (Figure 6a,b). Further FSP passes resulted in more homogeneous distribution of RPPs by the FSP track width in the stir zone (Figure 6e,f).

Judging by the results of macroscopic examination, the acceptable homogeneity of RPP distribution in the SZ is achieved after at least four FSP passes. Therefore, further microstructural studies were carried out on samples processed using four and eight (4 + 4) passes. The microstructures of the four-pass FSPed sample SZ are represented by equiaxed fine recrystallized AA5056 grains [24] and evenly distributed dark RPPs (Figure 7a). The thermomechanically affected zones (TMAZ) are on the periphery of the SZ and therefore the plasticized metal flow intensity there is not that high as in the SZ. There was such a difference in the results of accumulation of the RPPs (Figure 7b). The pin-driven and shoulder-driven metal flows may collide and form a stagnant zone in the subsurface layer, where RPPs also agglomerate in the form of a layer (Figure 7c) which would be a stress concentration zone during mechanical tests and thus increase the risk of cracking and failure of the material in this place.

After eight consecutive passes (Figure 8a–c) the distribution of RPs throughout the volume of the treated area becomes more uniform. The RPP agglomerates and the dense RPP interlayer were completely intermixed with the SZ metal (Figure 8b,c), while a fine-grained structure was formed in SZ with RPPs evenly distributed throughout the AA5056 matrix. According to the grain size distribution (Figure 9a–c) it can be noted that the mean size of grains formed in different zones after four-pass FSP is almost the same with greater scatter as approaching closer to the base metal. In case of eight FSP passes the mean grain sizes are somewhat decreased as compared to those after four passes (Figure 10a–c). In general, this difference is not very important. More important is the fact that grains did not grow after the multi-pass FSP.

To identify the phases obtained in each of the zones, the EDS elemental composition was determined in corresponding points of samples after four-pass (Table 1) and eight-pass experiments (Table 2). The elemental compositions in points 1 (Figure 11b–d) correspond to that of the base AA5056. The elemental composition of the RPP agglomerate (Figure 11b, pos. 2, 3, 4) approximately corresponds to that of the iron tungstate with some additions of admixed aluminum. Precipitates found in the SZ nugget and TMAZ contain already much more aluminum along with W and Zr (Figure 11c, pos. 2, 4) and (Figure 11d, pos. 2, 4), respectively. It may be suggested that these particles are at least partially composed of WAl_12_ and ZrAl_3_.

The EDS spectrum of particles in Figure 11c,d, pos. 3 contains less aluminum but more oxygen and zirconium and therefore may be identified as those containing ZrO_2_ and WO_3_.

The same studies were carried out on the 8-pass FSPed sample. It is clearly seen that the RPP agglomerate subsurface layer has disappeared (Figure 12a) and the corresponding SEM image allows observing the relatively homogeneous distribution in the matrix (Figure 12b, pos. 1) particles (Figure 12b, pos. 2, 3, 4) whose EDS spectra (Table 2) testify on the presence of either oxygen- or aluminum-rich particles. Almost the same results can be inferred from the EDS spectra obtained from the SZ nugget (Figure 11c) and TMAZ (Figure 12d).

The TEM images of particles distributed in the stir zone after eight FSP passes show that these particles are of irregular shape (Figure 13a) in contrast to rectangular as-received ZrW_2_O_8_ ones (Figure 2c)_._ The EDS elemental analysis shows them containing less than 10 at.% of W and less than 5 at.% of Zr (Figure 13b). At the same time, they contain more than 60 at.% of Al. It is interesting that particle 1 contains more W than the as-received ZrW_2_O_8_ (Figure 13b, probe zone 8) as well as higher amounts of magnesium, iron, and chromium. The latter two elements came from the FSP tool by adhesion transfer and formation of Al-Fe intermetallic compound on the steel tool [31]. Such a finding allows suggesting that these micron-sized particles were precipitated from the solid solution, which, in its turn was obtained by mechanical stirring of the particles in the plasticized aluminum alloy. Earlier we observed dissolution and precipitation of iron-containing so-called insoluble particles in FSW on aluminum alloys [24].

The TEM shows that the as-received ZrW_2_O_8_ particles suffered dramatic transformation after FSP in the aluminum alloy. First of all, they have lost their rectangular shape, became saturated with aluminum and acquired loose structure (Figure 14a,b). There is some transition from the particle core to the periphery, which may be formed by diffusion of its elements into aluminum alloy matrix. Such a mass transfer generates vacancies which then condense in dislocation loops around the particles (Figure 14b). First of all, aluminum serves to decompose the ZrW_2_O_8_ by depriving it of oxygen and immediately forming the Al_2_O_3_. The reduced W and Zr both took part in further evolution. The excess of aluminum results in forming also a W/AL intermetallic compound.

Figure 15a,b shows the XRD patterns obtained from the four-pass and eight-pass samples, respectively. The diffractograms allow identifying the presence of phases such as α-Al, ZrAl_3_, WAl_12_, β-ZrW_2_O_8_, t-ZrO_2_ and Al_2_O_3_. These XRD data serve to support the suggestions made after examining the EDS data, i.e., that four FSP passes is enough to transform the cubic α-ZrW_2_O_8_ (Figure 2a) into tetragonal β-ZrW_2_O_8._ Such a metastable tetragonal ZrW_2_O_8_ phase was discovered during the decomposition of ZrW_2_O_8_ at 1373° K [32]. Since it is known that the transformation of the α-ZrW_2_O_8_ at 155 °C into β-phase is reversible and cannot be identified from the diffractograms in Figure 15, we designated it β’-ZrW_2_O_8_. This phase may partially decompose into zirconium and tungsten oxides, which then become partially reduced by aluminum to free metals which then react with the remnant aluminum and from intermetallic WAl_12_ and ZrAl_3_ particles.

### 3.3. Microhardness Profiles of the FSP-ed RPPs/AA5056 Composites

The distribution of microhardness numbers across the SZ ion the 1-pass FSPed sample shows the considerable scatter that could be related to inhomogeneous distribution of RPPs and unreacted source powder agglomerates (Figure 16a). On the contrary, both four-pass and eight-pass FSPed samples demonstrate the SZ hardening by inhomogeneously distributed RPPs. (Figure 16b,c). The TMAZ metal has lower microhardness in both cases and the reason behind such a finding may be less content of reinforcing particles and more unreacted iron tungsten in these stagnant zones. The same situation was with FSW on AA2024 where precipitates grew larger in the TMAZ and therefore lost their reinforcing effect by overaging [33].

### 3.4. Ultimate Tensile Strength and the Engineering Strain of the FSP-ed RPPs/AA5056 Composites

To perform quasi-static tensile tests, the dog-bone samples with their tensile axes oriented along the direction perpendicular to the FSP track centerline (Figure 17a,b). The stress–strain curves show very low strength characteristics of the extremely inhomogeneous one-pass sample while both four-pass and eight-pass ones demonstrate their UTS close to that of the base hot-rolled metal but the enhanced ductility (Figure 17c).

At the same time, the fractographic analysis of the four-pass FSPed sample showed that large particulates are present on fracture surface (Figure 18a,b) with the background surface ductile fracture surface consisting of ridges and dimples (Figure 18c,d). The dimples were formed by pulling out ductile metal around the hard non-deformed particles. Those are RPPs almost homogeneously distributed in the AA5056 matrix.

A considerably lower number of large RPPs can be observed on the fracture surface of the eight-pass FSPed sample (Figure 19a) while the rest of it is occupied by ductile fracture surface (Figure 19b) with an even more homogeneous distribution of smaller RPPs in the ductile matrix (Figure 19c). The RPPs have good bonding with the matrix so that each of them is covered by the matrix material (Figure 19d). These samples demonstrated the intragranular ductile fracture mechanisms as well as good tensile strength and improved plasticity.

### 3.5. Tribological Behavior of the FSP-ed RPPs/AA5056 Composites

The dependence of the coefficient of friction of the sample on the test time is shown in Figure 20a,b. The coefficient of friction (CoF) dependencies on time shows the average CoF values for the four-pass and eight-pass samples are µ = 0.34 and µ = 0.32, respectively, as compared to analogous characteristics of the four-pass and eight-pass FSPed AA5056 samples, namely, µ = 0.54 and µ = 0.50, respectively. Therefore, in-situ FSP admixing the iron tungstate resulted in reducing the CoF values by 36–37% depending on the number of passes.

Measurement of wear by losses in height (Δ*H*) and mass (Δm) of the FSP-ed RPPs/AA5056 composites caused by sliding tests (Table 3) showed the decrease in wear rate depending on the number of passes. So, after the four successive passes, Δ*H* was 0.62 mm, and by the eighth pass it decreased by 0.12 mm. In this case, the wear rate decreased insignificantly.

The wear intensity was calculated using the formula as follows:$$I_{w}=\frac{\Delta H\;\cdot S}{\omega \cdot t}\cdot P;$$
where ΔH—height difference, m; S—cross section areas of the wear track, m^2^; ω—angular frequency, Rad/s; *t*—time, s.

The GIAXRD from the worn surfaces of the four-pass (Figure 21a) and eight-pass (Figure 21b) composites allowed for identifying the presence of iron oxide Fe_2_O_3_, in addition to the phases formed during the FSP (Figure 15a,b).

The SEM BSE images of worn surfaces obtained on both four- (Figure 22a,c) and eight-pass (Figure 22b,d) samples show that the predominant wear mechanism was the abrasive wear with the formation of grooves due to micro-ploughing. Delaminated layers in the form of craters indicate local adhesive wear. Local agglomerations of the oxidized wear products and abrasive particles can be observed (Figure 22c,d) that may then form mechanically mixed layers. Bright areas allow for observing concentrations of high atomic number elements such as tungsten and zirconium.

The EDS analysis of the friction surface (Figure 23 and Figure 24; Table 4) of the samples showed the presence of a large amount of oxygen on the worn surfaces, which, in combination with the XRD data (Figure 21), indicated on intense tribooxidation during sliding.

Figure 23 shows the SEM BSE images and combined EDS images on the worn surfaces of four-pass (Figure 23a,b) and eight-pass (Figure 23c,d) RPPs/AA5056 composites. It can be seen that there are bright and dark areas (Figure 23a,c), which can be interpreted as mechanically mixed layer formed by means of transfer of elements from the steel counterbody and heavily oxidized RPPs/AA5056 composite, respectively. The mechanically mixed layer is not practically oxidized in the presence of aluminum which takes the oxygen for forming more alumina (Figure 21).

There are bright particles on the worn surfaces that contain W, Zr and O (Figure 24; Table 4, pos. 1) and the same particles can be observed in Figure 24a, pos. 1′ directly below the worn surface in the bulk of the composite.

Subsurface section views demonstrated that the bight particles enriched with elements such as W, Zr and O can be severely deformed when found below the worn surface inside the subsurface deformed layers forming something similar to the quasi-viscous multi-layer subsurface structure (Figure 25, Table 5).

## 4. Discussion

The results of this work have shown that under the selected FSP parameters, the dense composite stir zones in-situ reinforced with intermetallic and oxide particles were formed. When FSP admixed into AA5056 matrix the α-ZrW_2_O_8_, rectangular high aspect ratio particles are transformed into β’-ZrW_2_O_8_ ones, which are metastable even at the room temperature. During the FSP, these metastable particles decompose into WO_3_ and ZrO_2_ oxides, which then are reduced in the presence of aluminum so that oxygen is redistributed into Al_2_O_3_. In the first place, it is subject to tungsten since it then reacts with aluminum to form WAl_12_. The ZrO_2_ dioxide reduction by aluminum is less intense so that some ZrO_2_ particles are left in the composite even after the eight-pass FSP. The reduced zirconium particles interact with aluminum to produce ZrAl_3_.

It was shown [16] that during sintering of a mixture of Al-ZrW_2_O_8_ powders at 600 °C in an argon atmosphere for 1 and 5 h, zirconium tungstate decomposed into its constituent oxides WO_3_ and ZrO_2_ with subsequent formation of intermetallic compounds WAl_12_ and ZrAl_3_. Our estimates of temperatures in the friction contact zone during the FSP on an aluminum alloy showed that the temperature does not exceed 400 °C and, at first glance, does not exceed the decomposition temperature of ZrW_2_O_8_ in the aluminum matrix (410 °C) [6,9,17]). At the same time, it is known [24] that such intense thermomechanical effects reduce the temperature threshold of high-temperature transformations due to the combined effect of temperature and shear stresses.

Currently, there is no information in the literature on the tribological characteristics of aluminum alloys reinforced with ZrW_2_O_8_. At the same time, it is known [34] that a significant positive thermal expansion is a critical problem in many tribological applications, especially when the material is subjected to intense heating. This is a particularly important factor for extremely localized deformations and energy dissipation, i.e., under conditions of sliding friction, when two rough surfaces are in contact with each other only by real contact areas, and intense heating leads to additional (thermally induced) stresses between the contacting and thermally expanding irregularities. These thermal stresses have at least two negative effects on sliding friction. Firstly, these stresses contribute to the achievement of local critical stress values and thereby contribute either to inelastic deformation or to the fracture of the asperities.

It was shown in [35,36] that the addition of second phases with negative or positive thermal expansion coefficients, but noticeably lower than those of the TiNi austenitic matrix, is an effective way to expand the temperature range in which wear losses are minimal. The greater the difference in thermal expansion coefficient between austenite TiNi and the added second phase, the greater the temperature range of low wear. In addition, the addition of the hard secondary phases reduced the overall wear loss of the TiNi/ZrW_2_O_8_/W composites by an order of magnitude.

Tribological adaptation mechanisms show the effect of tribo-oxidation of W and Fe contained in either one or both counterbody materials with in-situ formation of lubricative FeWO_4_ and Fe_2_WO_6_ mixed oxides [37,38,39]. For example, it was established that some quasi-viscous subsurface multilayer structures have been formed on the WC/Y–TZP–Al_2_O_3_/Hadfield steel composites rubbed against an HSS counterbody at high sliding speeds. These structures consisted of FeWO_4_ and Fe_2_WO_6_ which encapsulated wear debris and carbide fragments thus forming microcomposite mechanically mixed layers capable of healing the voids and cracks as well as simultaneously reducing wear and friction [38].

As shown by our results (Figure 20, Table 3), there is a clear improvement in the tribological characteristics of the studied composites prepared using the FSP. The reduced values of wear intensity and CoF can be logically attributed to the combined action of three factors: (1) reduced CTE values of near-surface regions (relative to CTE of the aluminum matrix ~22.4 × 10^−6^ K^−1^) due to the presence in them of a mixture of simple and complex oxides (including Al_2_O_3_, ZrO_2_, β-ZrW_2_O_8_) and intermetallic compounds (WAl_12_ and ZrAl_3_); (2) the reinforcing effect of well-mixed solid named oxides and intermetallic compounds; and (3) tribochemical adaptation.

Apparently, the FSP method used in this work made a great contribution to the formation of the second factor. It is known [24] that the FSP method for hybrid composites in situ has several advantages over other methods used, namely: (i) more thermodynamically stable matrix reinforcement, (ii) coherent/semi-coherent bonding at particle/matrix interfaces, and (iii) formation of smaller reinforcing particles evenly distributed in the matrix.

The multi-layer subsurface structures have been found in this investigation that revealed their quasi-viscous behavior in sliding (Figure 25). Therefore, the third factor of tribochemical adaptation may be very important from the viewpoint of tribological behavior of the RPPs/AA5056 composites. Along with that, the insufficient accuracy of the analytical methods used in this work did not allow us to unambiguously determine as to which one of the above-mentioned factors serves mainly to reduce wear and friction in this particular system. This task will be in focus of our future studies.

## 5. Conclusions

This work was devoted to studying the stability of zirconium tungstate in an aluminum alloy during FSP and potential use of the decomposition products for reinforcing the AA5056 stir zone. This approach was dictated by the necessity of developing novel aluminum alloy-base composites with lower thermal expansion as well as reinforced by intermetallic and oxide precipitates for better strength and wear resistance.

The results of this work have shown that the FSP parameters provided formation of dense composite surface layers. The FSP of α-ZrW_2_O_8_ in the plasticized AA5056 resulted in the high-temperature into α → β transformation, which then decomposed into its oxides WO_3_ and ZrO_2_. The next stage was formation of intermetallic compounds WAl_12_ and ZrAl_3_. The latter ones served for reinforcing the AA5056 stir zone in combination with the Hall–Petch mechanisms.

There is a clear improvement in the mechanical and tribological characteristics of the resulting composites. The reduced values of wear intensity and friction coefficient can be logically attributed to the combined action the Hall–Petch mechanism and due to the presence oxides Al_2_O_3_, ZrO_2_, β’-ZrW_2_O_8_ and intermetallic compounds WAl_12_ and ZrAl_3_.

The future efforts will be related with improving the ZrW_2_O_8_ stability in FSP and studying the effect of CTE on wear and friction of the composites. Potential applications of the above-discussed composites maybe related to their improved tribological characteristics, for example in aerospace and vehicle-building industries.

## Figures and Tables

**Figure 1 materials-16-00817-f001:**
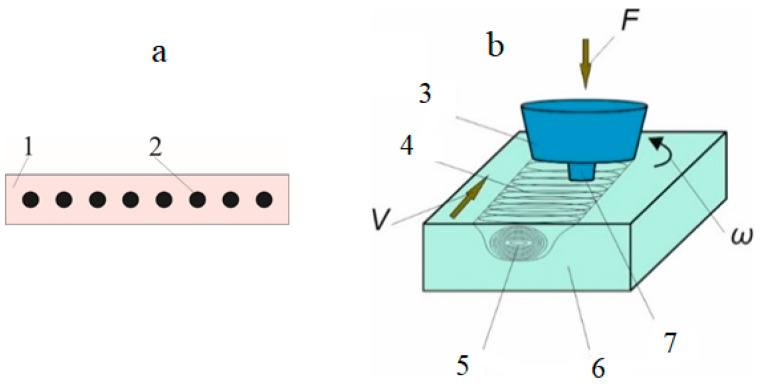
The AA5056 plate with drilled holes containing zirconium tungstate powder ZrW_2_O_8_ (**a**): 1—plate made of AA 5056 alloy; 2—holes loaded with the ZrW_2_O_8_ powder. The diagram of friction stir processing (**b**): 3—tool support shoulder; 4, 5—SZ; 6—base metal; 7—FSP tool’s pin; F is the applied load, N; V—tool traverse movement speed, mm/min; ω—tool rotation frequency, RPM.

**Figure 2 materials-16-00817-f002:**
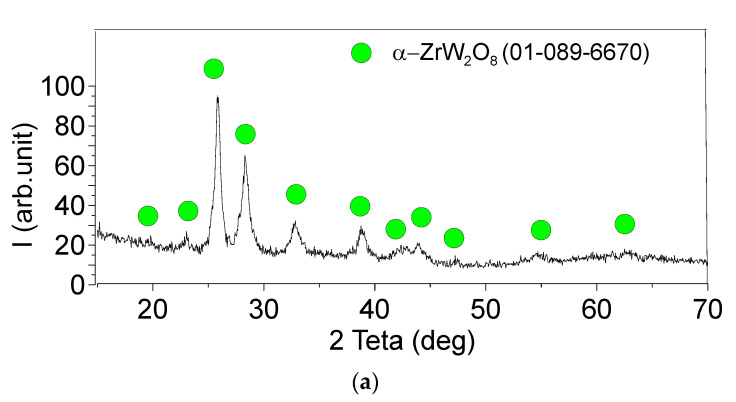
The XRD pattern (**a**), SEM (**b**) and TEM (**c**) images of as-received zirconium tungstate particles.

**Figure 3 materials-16-00817-f003:**
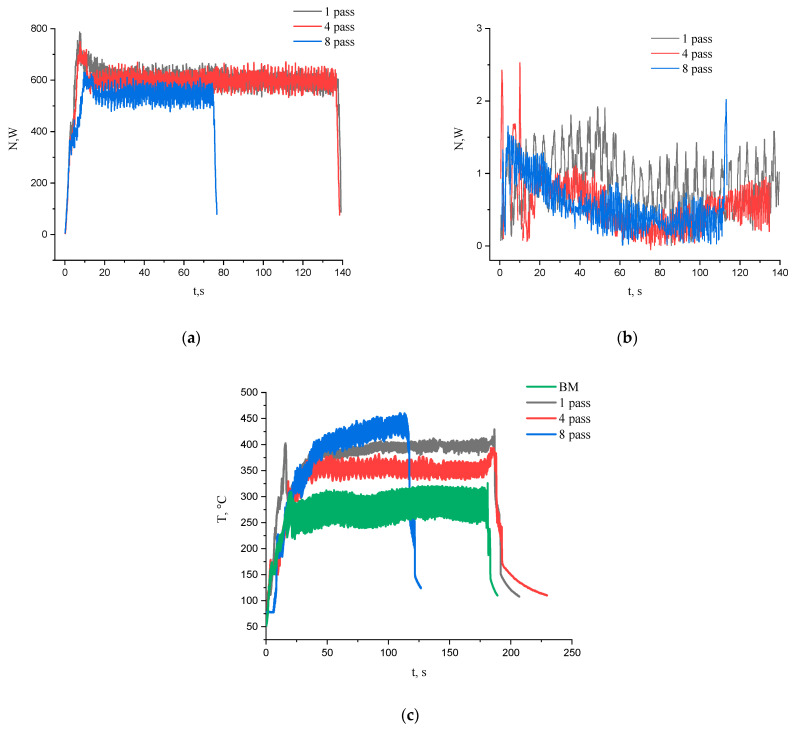
Torque power (**a**), reaction force power (**b**) and surface temperature (**c**) dependencies vs. FSP pass times.

**Figure 4 materials-16-00817-f004:**
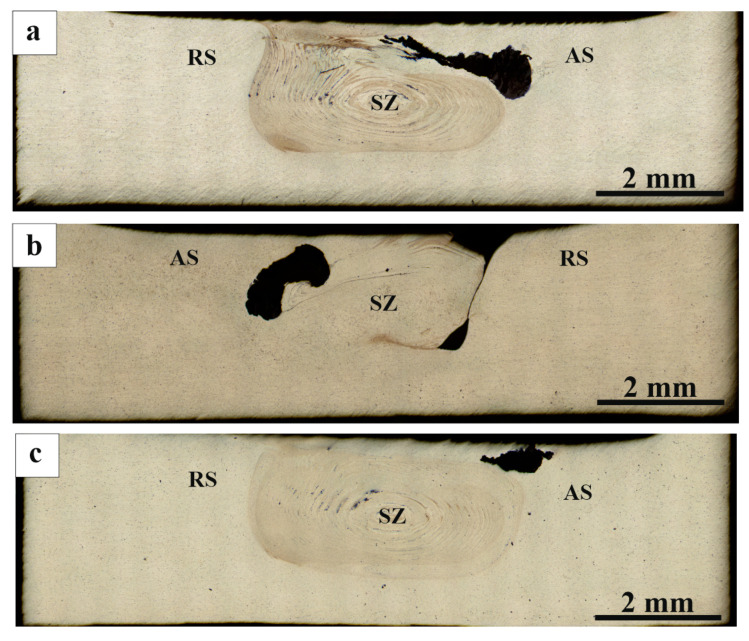
Macrostructure of the RPPs/AA5056 composite after: (**a**) 1st; (**b**) 2nd; (**c**) 3rd; and (**d**) 4th tool passes along the machining line. SZ—stir zone; RS—retreating side; AS—advancing side.

**Figure 5 materials-16-00817-f005:**
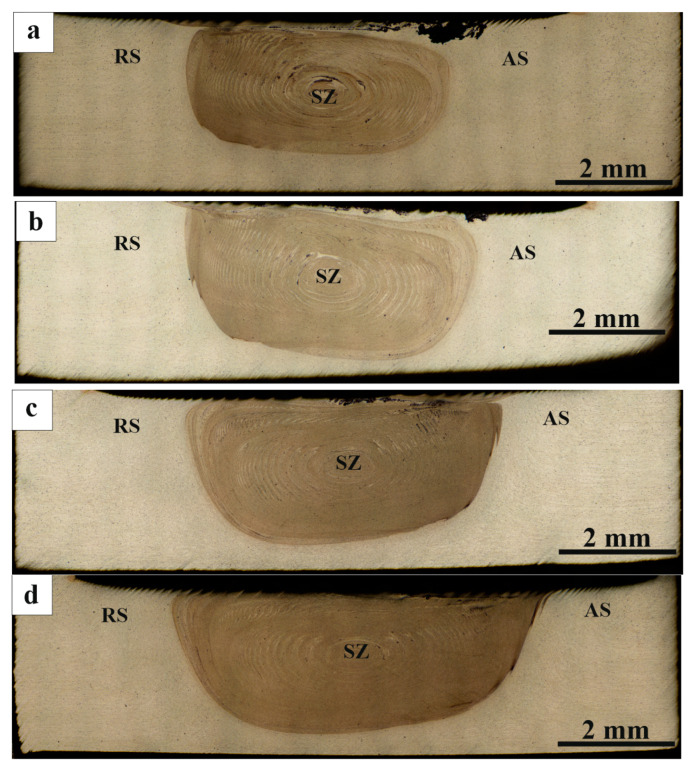
Macrostructure of RPPs/AA5056 composite after: (**a**) 1st; (**b**) 2nd; (**c**) 3rd; and (**d**) 4th additional tool passes along the machining line. SZ—stir zone; RS—retreating side; AS—advancing side.

**Figure 6 materials-16-00817-f006:**
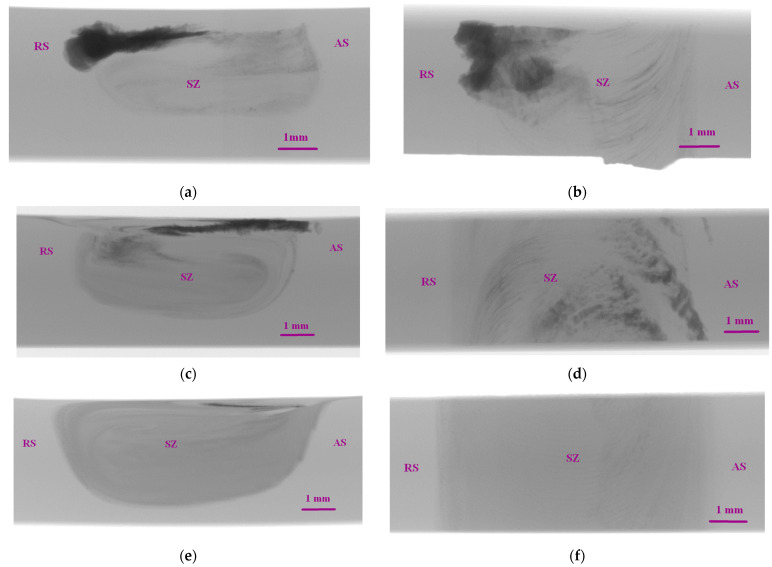
X-ray computer tomography images of the RPPs/AA5056 composite after: (**a**,**b**) 1st; (**c**,**d**) 4th; (**e**,**f**) 4th additional tool passes along the machining line. SZ—stir zone; RS—retreating side; AS—advancing side. (**a**,**c**,**e**) show the XCT layers as sectioned by the plane perpendicular to the FSP track axis. (**b**,**d**,**f**) show the XCT layer as seen from the direction perpendicular to the FSP track’s top surface.

**Figure 7 materials-16-00817-f007:**
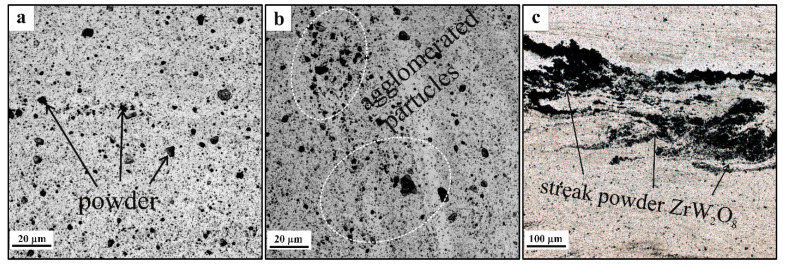
Microstructure of the FSP-ed RPPs/AA5056 composites after 4 passes: (**a**) stir zone, (**b**) thermomechanically affected zone; (**c**) subsurface RPP accumulation layer.

**Figure 8 materials-16-00817-f008:**
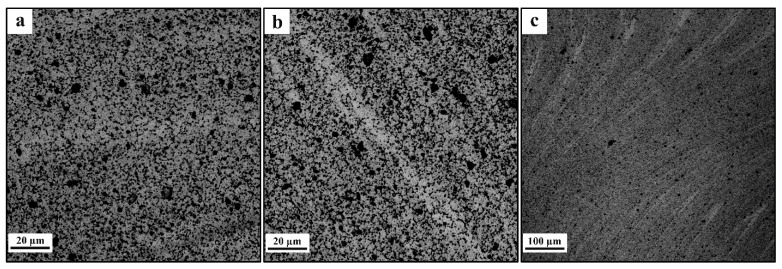
Microstructure of FSP-ed RPPs/AA5056 composite after 8 passes: (**a**) stir zone; (**b**) thermomechanically affected zone; (**c**) zone accumulations of RPs.

**Figure 9 materials-16-00817-f009:**
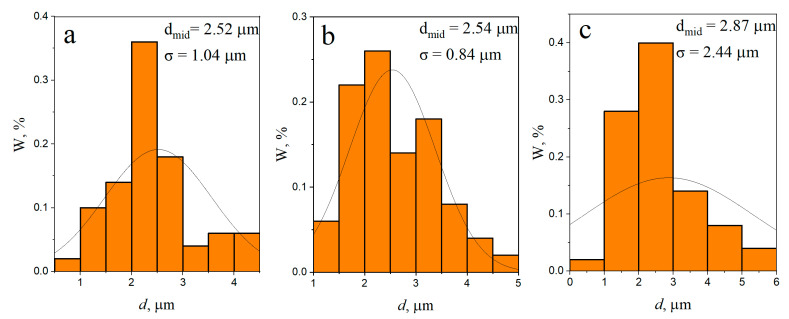
Average grain size and grain size distribution FSP-ed RPPs/AA5056 composite in the processing zone after 4 passes: (**a**) stir zone; (**b**) thermomechanically affected zone; (**c**) base metal.

**Figure 10 materials-16-00817-f010:**
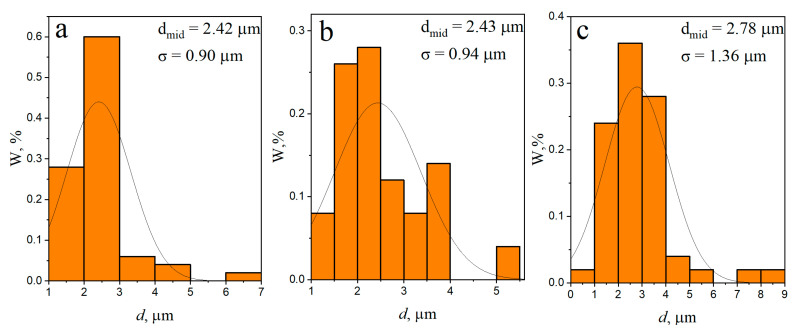
Average grain size and grain size distribution FSP-ed RPPs/AA5056 composite in the processing zone after 8 passes: (**a**) stir zone; (**b**) thermomechanically affected zone; (**c**) base metal.

**Figure 11 materials-16-00817-f011:**
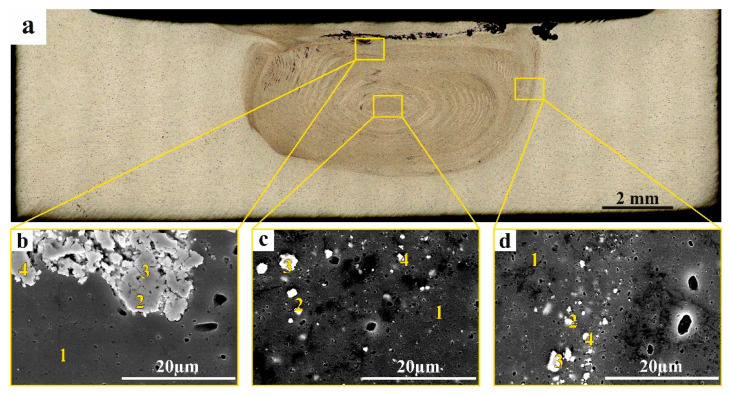
Macro (**a**) and micro (**b**–**d**) SEM graphs of FSP-ed RPPs/AA 5056 composite: (**a**) macrograph section of FSP-ed RPPs/AA5056 composite after 4 consecutive passes; (**b**) RPP accumulation zone; (**c**) stir nugget zone; (**d**) thermomechanically affected zone. Numbers on (**b**–**d**) indicate probe zones for which EDS elemental concentrations were determined indicated in Table 1.

**Figure 12 materials-16-00817-f012:**
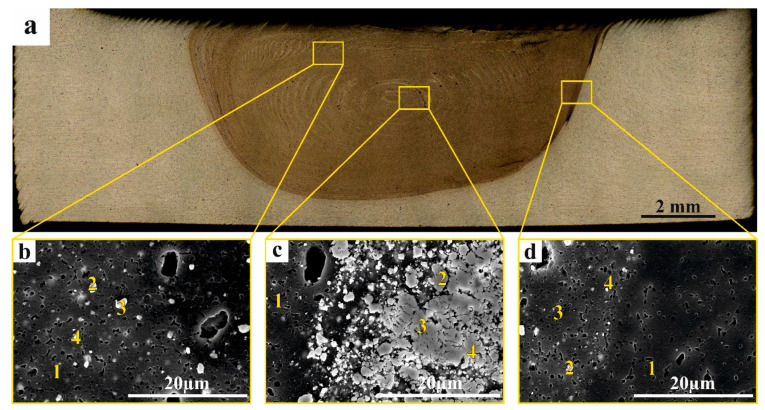
Macro (**a**) and micro (**b**–**d**) photos of FSP-ed RPPs/AA 5056 composite obtained by SEM: (**a**) macrograph section of FSP-ed RPPs/AA5056 composite after 8 consecutive passes with the tool; (**b**) RPP accumulation zone; (**c**) stir zone; (**d**) thermomechanically affected zone. Numbers on (**b**–**d**) indicate probe zones for which EDS elemental concentrations were determined indicated in Table 2.

**Figure 13 materials-16-00817-f013:**
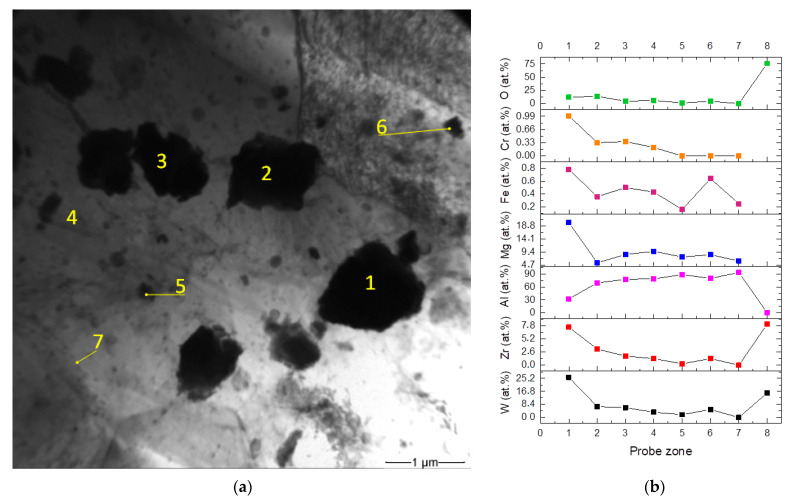
TEM image of stir zone metal after 8-pass FSP (**a**) and EDS elemental concentrations (**b**) in probe zones indicated in (**a**). Probe zone 8 is shown to reproduce the composition of the as-received ZrW_2_O_8_.

**Figure 14 materials-16-00817-f014:**
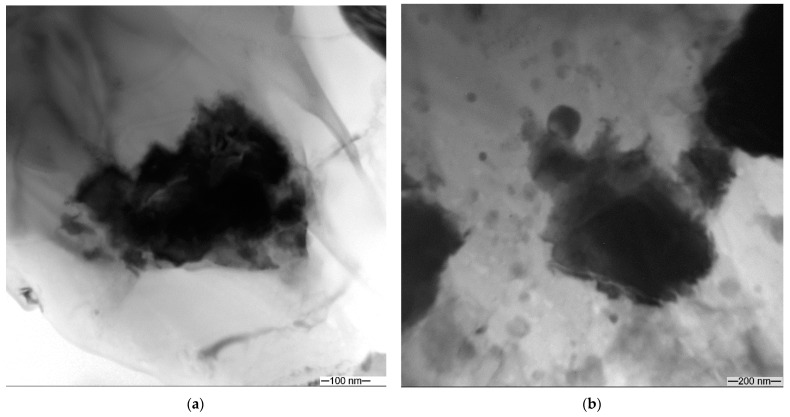
TEM image of particle resulted from decomposition of ZrW_2_O_8_ (**a**) and vacancy condensation loops around such a particle (**b**).

**Figure 15 materials-16-00817-f015:**
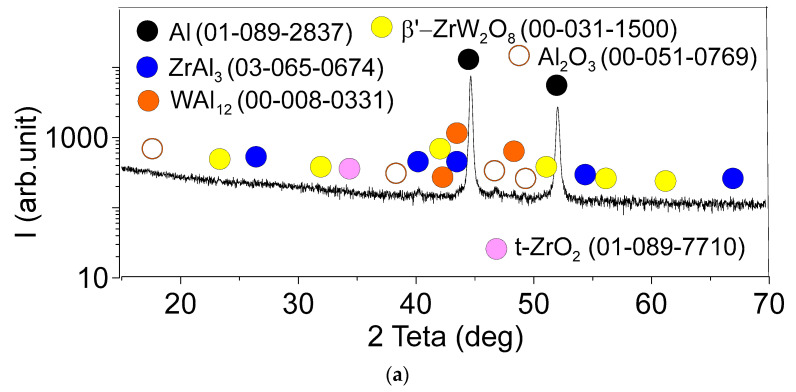
The XRD patterns of FSP-ed RPPs/AA 5056 composite after 4 (**a**) and 8 (**b**) passes.

**Figure 16 materials-16-00817-f016:**
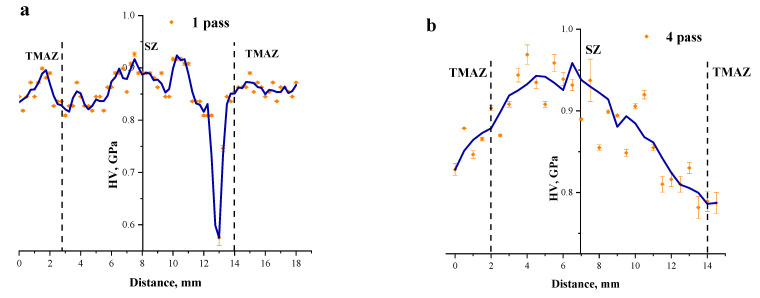
Microhardness profiles from the FSP-ed RPPs/AA5056 composite after: 1 pass (**a**); 4 passes (**b**) and 8 passes (**c**). SZ—stir zone; TMAZ —thermomechanically affected zone.

**Figure 17 materials-16-00817-f017:**
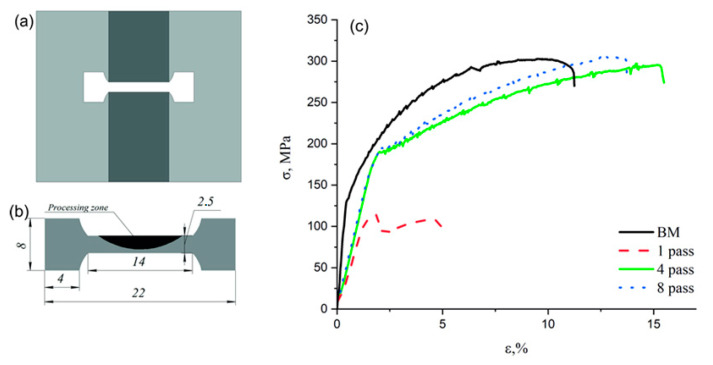
FSP-ed RPPs/AA5056 sample (**a**), tensile test specimen (**b**) and stress–strain curves obtained under quasi-static tension conditions (**c**).

**Figure 18 materials-16-00817-f018:**
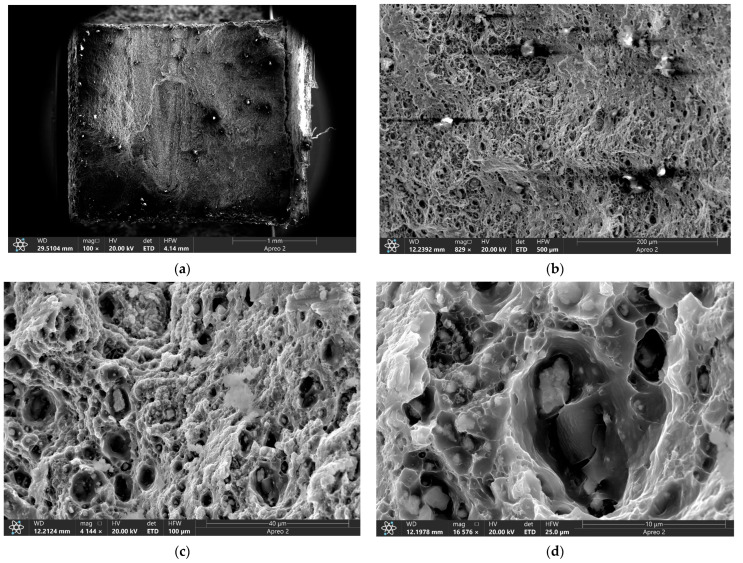
SEM BSE images of the tensile test fracture surfaces of the FSP-ed RPPs/AA5056 composite after 4 passes (**a**–**d**).

**Figure 19 materials-16-00817-f019:**
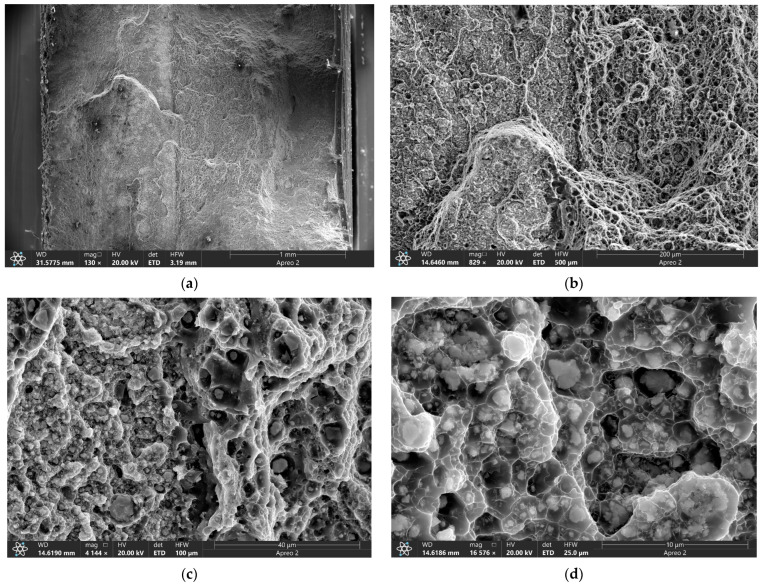
SEM BSE images of the tensile test fracture surfaces of the FSP-ed RPPs/AA5056 composite after 8 passes (**a**–**d**).

**Figure 20 materials-16-00817-f020:**
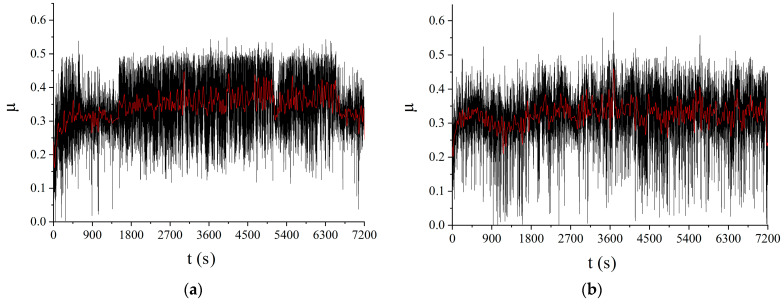
Coefficient of friction vs. time dependencies of FSP-ed RPPs/AA5056 composite: (**a**) FSP-ed RPPs/AA5056 composite after 4 passes; (**b**) FSP-ed RPPs/AA5056 composite after 8 passes.

**Figure 21 materials-16-00817-f021:**
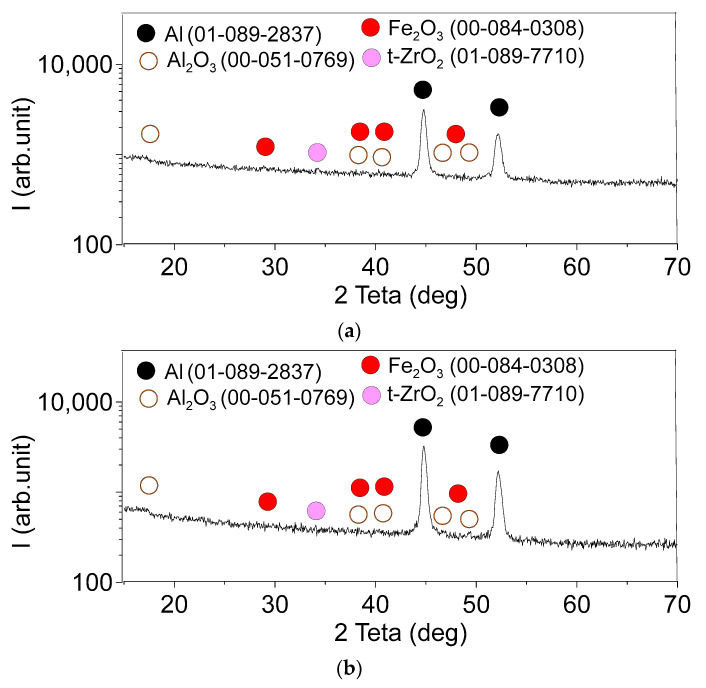
Glancing X-ray diffractograms with the incidence angle of 10° from the worn surface of FSP-ed RPPs/AA5056 composite after: (**a**) 4 passes and (**b**) 8 passes.

**Figure 22 materials-16-00817-f022:**
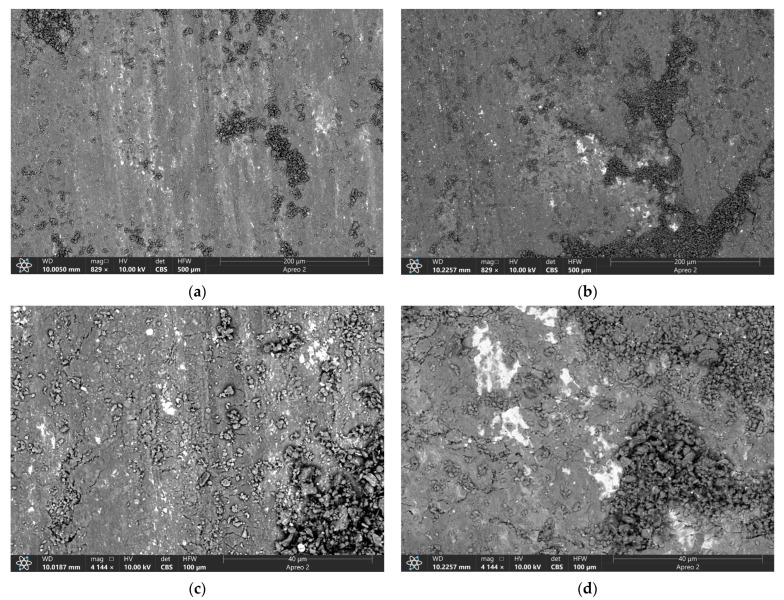
SEM BSE worn surface images of FSP-ed RPPs/AA5056 composite after 4 (**a**,**c**) and 8 (**b**,**d**) passes.

**Figure 23 materials-16-00817-f023:**
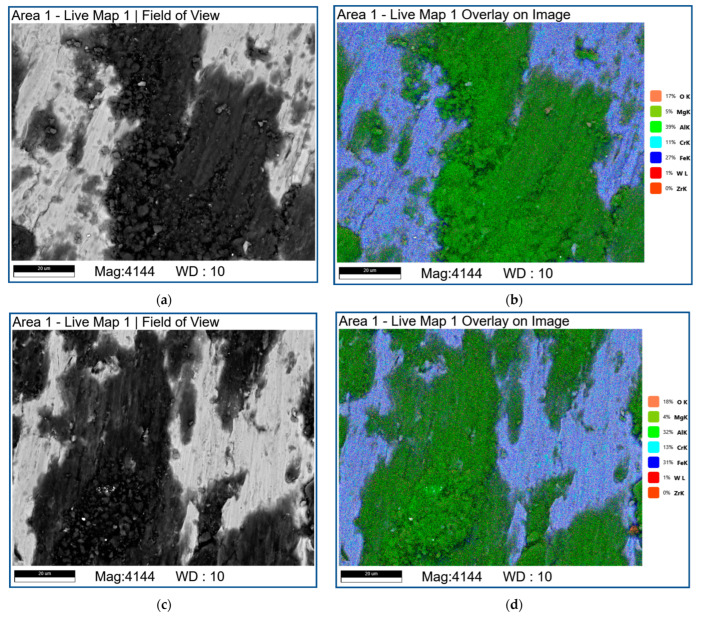
SEM BSE image (**a**,**c**) and combined EDS images (**b**,**d**) on the worn surface of FSP-ed RPPs/AA5056 composite after: 4 passes (**a**,**b**); 8 passes (**c**,**d**).

**Figure 24 materials-16-00817-f024:**
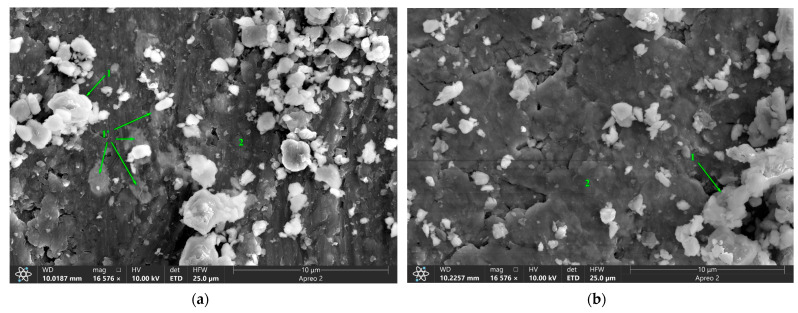
SEM images of worn surfaces of FSP-ed RPPs/AA5056 composites after: (**a**) 4 and (**b**) 8 passes. Numbers on (**a**,**b**) indicate probe zones for which EDS elemental concentrations were determined indicated in Table 4.

**Figure 25 materials-16-00817-f025:**
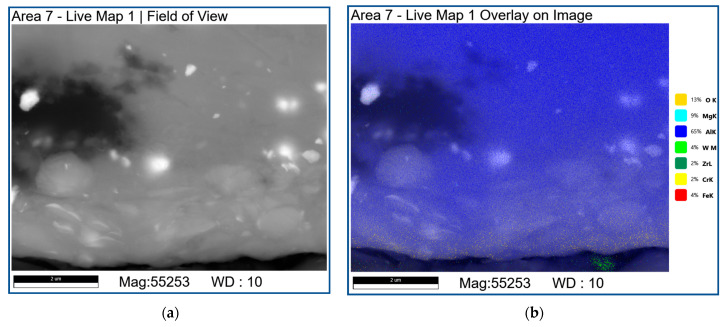
The SEM BSE images (**a**,**c**,**d**) and layered EDS images (**d**) of microstructures formed by sliding below the worn surface of FSP-ed RPPs/AA5056 composites after 8 passes. Numbers on (**d**) indicate probe zones for which EDS elemental concentrations were determined indicated in Table 5. The white arrow on (**d**) shows the direction of sliding.

**Table 1 materials-16-00817-t001:** Chemical composition (EDS) of an etched section of FSP-ed RPPs/AA 5056 after 4 tool passes.

Area	Point Number	Element, at.%				
		O	Al	Mg	W	Zr
RPPs accumulation zone	1	1.7	90.3	7.8	0.2	0.0
	2	68.2	1.4	0.0	12.8	17.6
	3	69.2	1.0	0.0	13.3	16.5
	4	70.7	1.4	0.0	12.5	15.4
SZ	1	1.0	90.7	8.2	0.1	0.0
	2	24.7	51.4	8.8	7.0	8.1
	3	66.7	8.6	0.5	14.1	10.1
	4	40.5	33.0	8.6	6.7	11.2
TMAZ	1	0.3	91.6	8.1	0.0	0.0
	2	24.5	55.5	6.5	6.4	7.1
	3	68.7	4.5	0.3	12.2	14.3
	4	40.2	36.5	4.5	6.6	12.2

**Table 2 materials-16-00817-t002:** Chemical composition (EDS) of an etched section of FSP-ed RPPs/AA 5056 after 8 tool passes.

Area	Point Number	Element, at.%				
		O	Al	Mg	W	Zr
RPPs accumulation zone	1	0.3	87.1	12.5	0.1	0.0
	2	45.0	32.5	3.5	7.7	11.3
	3	54.1	17.5	1.9	10.0	16.5
	4	20.9	58.0	7.7	6.2	7.2
SZ	1	1.7	90.1	8.0	0.2	0.0
	2	62.9	3.9	0.2	13.2	19.8
	3	69.2	3.5	0.2	14.0	13.1
	4	51.9	25.5	1.1	9.2	12.3
TMAZ	1	0.3	91.2	8.5	0.0	0.0
	2	20.9	58.0	7.7	6.2	7.2
	3	34.5	53.9	4.8	6.8	0.0
	4	66.0	10.6	2.4	7.7	13.3

**Table 3 materials-16-00817-t003:** The sliding wear test parameters and wear characteristics.

Sample	Counterbody	Rwear Track, mm	P, H	t, min	ω, RPM	Hs, mm	H_f_, mm	Δ*H*, mm	*I_w_*, mm^3^/m
passes AA5056	40 × 13	10	12	120	250	9.30	8.35	0.95	9.9 × 10^−3^
AA5056 + ZrW_2_O_8_ 4 passes	40 × 13	10	12	120	250	9.27	8.65	0.62	6.5 × 10^−3^
AA5056 + ZrW_2_O_8_ 8 passes	40 × 13	10	12	120	250	9.37	8.87	0.50	5.2 × 10^−3^

**Table 4 materials-16-00817-t004:** Chemical composition (EDS) of the worn surfaces of FSP-ed RPPs/AA5056 composites after 4 and 8 passes (Figure 24).

Sample	Point Number	Element, at. %						
		O	Al	Mg	Cr	Fe	W	Zr
4 passes	1	56.5	22.7	3.2	0.1	0.5	11.7	5.3
	2	44.7	49.0	4.1	0.3	1.6	0.1	0.1
8 passes	1	31.9	48.2	3.9	0.1	0.3	10.5	5.1
	2	6.6	84.5	6.8	0.3	1.7	0.1	0.0

**Table 5 materials-16-00817-t005:** Chemical composition (EDS) of the of microstructures formed by sliding below the worn surface of FSP-ed RPPs/AA5056 composites after 8 passes (Figure 25).

Sample	Point Number	Element, at. %						
		O	Al	Mg	Cr	Fe	W	Zr
8 passes	1	19.4	69.7	6.3	0.5	1.4	2.1	0.8
	2	20.1	69.3	6.9	0.3	1.2	1.6	0.7
	3	31.6	56.7	5.5	1.0	4.6	0.5	0.2
	4	28.9	60.7	5.3	0.8	3.9	0.3	0.2
	5	6.9	85.9	6.3	0.1	0.5	0.3	0.0

## Data Availability

Not applicable.

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
