# Peer review of "In-Situ Al-Mg Alloy Base Composite Reinforced by Oxides and Intermetallic Compounds Resulted from Decomposition of ZrW2O8 during Multipass Friction Stir Processing"

_materials, 2023, doi:10.3390/ma16020817_

Round 1
Reviewer 1 Report
The work of this paper is more detailed and has a large amount of data. It is recommended to receive and publish after minor modifications. What is the abbreviation of CTE? Please give the full name in detail. The diffraction spots in Fig. 2c are completely invisible. How is the resistance force power measured in the Fig. 3a and b?
Author Response
The work of this paper is more detailed and has a large amount of data. It is recommended to receive and publish after minor modifications. What is the abbreviation of CTE? Please give the full name in detail. The diffraction spots in Fig. 2c are completely invisible. How is the resistance force power measured in the Fig. 3a and b?
A: Thank you. The CTE stands for coefficient of thermal expansion, the corresponding explanation has been added. The reaction force was measured using a gauge fixed on the FSP tool shaft. The SAED pattern in Fig.2c has been improved for better visibility.

Reviewer 2 Report
Review Report
This study “ZrW2O8/AA5056 Al-Mg Alloy Surface Composites Fabricated by Friction Stir Processing” was aimed at studying the microstructure, mechanical and tribological properties of the AA5056 Al-Mg alloy treated with ZrW2O8 by friction mixing processing. The work has great potential to the research world. However, I recommend the following corrections before it is accepted for publishing.
Abstract
I suggest that recommendation of the composite’s area of application should be captured in the abstract.
Ln 20 pp.1, there was an omission of “of”.
Discussion of Results
For more clarity and easier comparison of microstructure, magnification of the SEM in Figures 6 and 7 should be the same. Make all the Figures 20 μm or 100 μm. This will make the discussion more understandable.
Ln 291 – 294 pp.13, Recast the whole sentence for more clarity.
Ln 296 pp 13, Chrome or chromium? Secondly, what is the source of this element?
Ln 296 – 299 pp 13, Many spelling errors. Correct.
In Ln 310, you spelled Al “aluminum”, but in Ln 312, it was spelled “aluminium”. Try to be be consistent with one throughout the work.
In the X-ray micrograph of Figure 14, what caused the loss of peak intensity to all other available phases except Al? Have they all turned to amorphous structure?
Ln 406 pp. 20, cite the source of that equation used.
Conclusion
Incorporate potential application of the developed composite.
Author Response
I suggest that recommendation of the composite’s area of application should be captured in the abstract.
A: Added
Ln 20 pp.1, there was an omission of “of”.
A: Thank you.
Discussion of Results
For more clarity and easier comparison of microstructure, magnification of the SEM in Figures 6 and 7 should be the same. Make all the Figures 20 μm or 100 μm. This will make the discussion more understandable.
A: We believe that the same magnifications used for Figures 7 and 8 (former Figures 6 and 7) may be justified because images Figure 7a,b and Figure 8a,b serve for comparison of the microstructures in the SZ and TMAZ obtained by the 4-pass and 8-pass FSP.
Figure 7с and Figure 8с demonstrate the subsurface layer with accumulated α-ZrW2О8 reaction product particles (RPPs). In other words, those relatively inhomogeneous formations are too large to show them as a whole agglomeration formed let’s say after 4-pass FSP. Therefore, we need low magnification in Figure 7c, and consequently in Figure 8c with more homogeneous distribution of RPPs after 8-pass FSP
Ln 291 – 294 pp.13, Recast the whole sentence for more clarity.
A: Thank you. The sentence was revised to read: “TEM images of particles distributed in the stir zone after 8 FSP passes show that these particles are of irregular shape (Figure 12a) in contrast to rectangular as-received ZrW2О8 ones (Figure 2c). The EDS elemental analysis shows them containing less than 10 at.% of W and less than 5 at.% of Zr (Figure 12b).”
Ln 296 pp 13, Chrome or chromium? Secondly, what is the source of this element?
A: Thank you. Corrected to read :” …chromium,…”. Chromium comes from the AISI 420 steel counterbody by means of adhesive transfer and mechanical intermixing with the rest of the wear debris. Some small amount may also come from the H13 FSP tool as a result of diffusion reaction with aluminum [see Ref.31].
Ln 296 – 299 pp 13, Many spelling errors. Correct.
A: Thank you. Corrected
In Ln 310, you spelled Al “aluminum”, but in Ln 312, it was spelled “aluminium”. Try to be be consistent with one throughout the work.
A: Thank you. Corrected
In the X-ray micrograph of Figure 14, what caused the loss of peak intensity to all other available phases except Al? Have they all turned to amorphous structure?
A: All XRD reflections from small intermetallic and oxide precipitates were shielded by the aluminum matrix because of very small contents of them. It is usual situation with the XRD detecting small amounts of phases in an abundant intermetallic matrix
Ln 406 pp. 20, cite the source of that equation used.
A: This is a simple formula that can be obtained directly from physics and in fact represents commonly used formula for calculating the wear speed combined with calculation of angular velocity from rotation frequency.
Conclusion
Incorporate potential application of the developed composite.
A: Added

Reviewer 3 Report
This research investigates the effect of zirconium tungstate ZrW2O8 powder on the microstructure, mechanical, and tribological properties of the AA5056 Al-Mg alloy treated by friction mixing processing. There are several serious issues in this manuscript that should be addressed:
· The writing of the manuscript needs much improvement. The manuscript needs to be fully revised since there are many typo errors and grammatical mistakes in the writing. As few examples, the manuscript should be revised as follows:
Line 110: “radiaton “ should be “radiation”
Line # 144: “…judged of using…” should be “…judged by using…”
Line 292: “…shape (Figure 12a). and do …” The full stop in the middle of the sentence should be removed.
Line # 293: less 10 at. % of W and less 5 at.% of Zr: Does it means: less than 10% of W and less than 5% of Zr?
· There is a need to explain how the process parameter’s values have been determined for this study.
· What is ZP in Tables 1 and 2? It has not been defined; does it refer to SZ??
· There is a need to improve the figures’ quality since most of the figures have very low quality. The legends, scale bars, and details in many figures are hard to read. Axes scales and legends should have a consistent font size in all figures.
· In line #53, different methods to create aluminum composites have been listed. Two important solid-state metal deposition techniques which have not been listed are “Friction Surfacing” and “Lateral Friction Surfacing.” They have been used or have many potentials for the deposition of aluminum alloys mixed with reinforcing particles. There is a need to explain these two solid-state deposition techniques and cite appropriate references to introduce these approaches.
· The conclusion of the study should be rewritten. The conclusion should restate the purpose and the importance of the study, the process, the most important findings of the study, and an overview of future research possibilities.
Author Response
Line 110: “radiaton “ should be “radiation”
A: Thank you. Corrected
Line # 144: “…judged of using…” should be “…judged by using…”
A: The sentence was revised to read:” The FSP process intensity can be evaluated by such parameters as torque of tool rotation, mechanical resistance force and surface temperatures”.
Line 292: “…shape (Figure 12a). and do …” The full stop in the middle of the sentence should be removed.
A:The sentence was revised to read:” TEM images of particles distributed in the stir zone after 8 FSP passes show that these particles are of irregular shape (Figure 12a) in contrast to rectangular as-received ZrW2О8 ones (Figure 2c)”.
Line # 293: less 10 at. % of W and less 5 at.% of Zr: Does it means: less than 10% of W and less than 5% of Zr?
A: the sentence was revised to read: “The EDS elemental analysis shows them containing less than 10 at.% of W and less than 5 at.% of Zr (Figure 12b)”.
There is a need to explain how the process parameter’s values have been determined for this study.
A: This information has been added to the Section 2 Materials and Methods as follows: “The choice of the FSP parameters was based on our previous experimenting with friction stir welding and processing on the AA5056. This set of parameters allowed obtaining defectless stir zones and micron-sized grains that provided higher than 100% welded joint strength in comparison with that of the as-received hot-rolled AA5056”.
What is ZP in Tables 1 and 2? It has not been defined; does it refer to SZ??
A: Thank you. Revised to read: “SZ”
There is a need to improve the figures’ quality since most of the figures have very low quality. The legends, scale bars, and details in many figures are hard to read. Axes scales and legends should have a consistent font size in all figures.
A: Thank you.
In line #53, different methods to create aluminum composites have been listed. Two important solid-state metal deposition techniques which have not been listed are “Friction Surfacing” and “Lateral Friction Surfacing.” They have been used or have many potentials for the deposition of aluminum alloys mixed with reinforcing particles. There is a need to explain these two solid-state deposition techniques and cite appropriate references to introduce these approaches.
A: Thank you. The references have been added with the corresponding discussion of the methods to the Introduction section.
The conclusion of the study should be rewritten. The conclusion should restate the purpose and the importance of the study, the process, the most important findings of the study, and an overview of future research possibilities.
A: Thank you. The missing points have been added
